# Morphological, Anatomical, and Phytochemical Studies of *Carlina acaulis* L. Cypsela

**DOI:** 10.3390/ijms21239230

**Published:** 2020-12-03

**Authors:** Maciej Strzemski, Bartosz J. Płachno, Barbara Mazurek, Weronika Kozłowska, Ireneusz Sowa, Krzysztof Lustofin, Daniel Załuski, Łukasz Rydzik, Dariusz Szczepanek, Jan Sawicki, Magdalena Wójciak

**Affiliations:** 1Department of Analytical Chemistry, Medical University of Lublin, 20-093 Lublin, Poland; i.sowa@umlub.pl (I.S.); 91chem91@gmail.com (J.S.); kosiorma@wp.pl (M.W.); 2Department of Plant Cytology and Embryology, Institute of Botany, Faculty of Biology, Jagiellonian University in Kraków, 30-387 Krakow, Poland; krzysztof.lustofin@doctoral.uj.edu.pl; 3Analytical Department, New Chemical Syntheses Institute, 24-110 Puławy, Poland; barbara.mazurek@ins.lukasiewicz.gov.pl; 4Department of Pharmaceutical Biology, Wroclaw Medical University, 50-556 Wroclaw, Poland; weronika.kozlowska@umed.wroc.pl; 5Department of Pharmaceutical Botany and Pharmacognosy, Ludwik Rydygier Collegium Medicum, Nicolaus Copernicus University, 85-094 Bydgoszcz, Poland; daniel_zaluski@onet.eu; 6Faculty of Physical Education and Sport, Institute of Sport, University of Physical Education in Krakow, 31-541 Kraków, Poland; lukasz.gne@op.pl; 7Chair and Department of Neurosurgery and Paediatric Neurosurgery, Medical University of Lublin, 20-090 Lublin, Poland; dariusz.szczepanek@umlub.pl

**Keywords:** essential amino acid, unsaturated fatty acids, chlorogenic acids, tocopherols, mineral composition

## Abstract

*Carlina acaulis* L. has a long tradition of use in folk medicine. The chemical composition of the roots and green parts of the plant is quite well known. There is the lowest amount of data on the cypsela (fruit) of this plant. In this study, the microscopic structures and the chemical composition of the cypsela were investigated. Preliminary cytochemical studies of the structure of the *Carlina acaulis* L. cypsela showed the presence of substantial amounts of protein and lipophilic substances. The chemical composition of the cypsela was investigated using spectrophotometry, gas chromatography with mass spectrometry, and high-performance liquid chromatography with spectrophotometric and fluorescence detection. The cypsela has been shown to be a rich source of macro- and microelements, vegetable oil (25%), α-tocopherol (approx. 2 g/kg of oil), protein (approx. 36% seed weight), and chlorogenic acids (approx. 22 g/kg seed weight). It also contains a complex set of volatile compounds. The *C. acaulis* cypsela is, therefore, a valuable source of nutrients and bioactive substances.

## 1. Introduction

Seeds of many plant species are a valuable source of various nutrients, e.g., oil, protein, carbohydrates, and biologically active secondary metabolites. Seed oils are rich in unsaturated fatty acids [1], which are highly important for human health, as they prevent the development of many diseases such as cancer, cardiovascular diseases, and diabetes. They are also essential for the proper development of the fetus and cognitive functions [2]. Tocols (tocopherols and tocotrienols) present in oils, referred to as vitamin E, play important roles in the human organism, preventing oxidation of the components of the cell membrane, polyunsaturated fatty acids, and low-density lipoproteins [3]. Additionally, seeds have a diversified mineral composition with high content of various micro- and macroelements [4]. Plant seeds are, therefore, a valuable source of phytochemicals with great medicinal and dietary importance. They are also important in animal nutrition. Moreover, seed oils are a desirable component of various cosmetic preparations [5,6]. This prompts research on plant seeds that have not been studied so far.

*Carlina acaulis* L. (Figure 1A) is a monocarpic perennial plant from the Asteraceae family occurring in South and Central Europe. *Carlina* root has been used in medicine since ancient times mainly for treatment of skin diseases, in treatment of dental and gastrointestinal tract diseases, and as an anthelmintic and diuretic agent. Some reports also indicate the therapeutic potential of aerial parts in skin diseases [7,8]. The chemical composition of the roots and green parts of *C. acaulis* is quite well known and established mostly by chromatographic techniques: high-performance thin-layer chromatography (HPTLC), high-performance liquid chromatography (HPLC), and gas chromatography (GC). For example, carlina oxide and inulin were found as the main phytochemicals in roots [9], and leaves were shown to contain flavonoids [10], chlorogenic acids, and considerable amounts of oleanolic and ursolic acids [11,12]. In contrast, there are scarce reports on the phytochemistry of seeds. They are limited to one publication from 1969 [13], in which the fatty acid profile of *C. acaulis* fruit oil was examined and only five fatty acids were identified and quantified.

The seeds of Asteraceae plants are botanically their fruits and are referred to as cypselae. The cypsela is a complex, indehiscent, dry, unilocular fruit with a single seed not adnate to the pericarp and derived from an inferior ovary [14]. The data in the literature indicate that cypselae of some Asteraceae species are rich in protein and oil (e.g., [15]), and immatured cypselae of some genera (*Taraxacum*, *Hieracium*, *Pilosella*) contain large amounts of polysaccharides [16,17,18]. 

The aim of our study was to investigate the morphological and anatomical structure of *C. acaulis* cypselae using such imaging techniques as light and scanning electron microscopy (SEM). Moreover, the chemical composition of cypselae was evaluated taking into account the content of primary and secondary metabolites as well as the mineral profile. The following techniques have been used: gas chromatography with mass spectrometry (GC-MS), HPLC with photodiode (DAD) and fluorescence (FLD) detection, ultra-performance liquid chromatography with fluorescence detection (UPLC-FLD), inductively coupled plasma-optical emission spectrometry (ICP-OES), attenuated total reflectance infrared spectroscopy (ATR-IR), and UV-VIS spectrophotometry. To our knowledge, this is the first study on the cell structure and chemical composition of the *C. acaulis* cypsela.

## 2. Results and Discussion

### 2.1. Macro- and Microscopy Analysis

*C. acaulis* cypselae (Figure 1B) are oblong, with 4–6 mm length and approx. 1 mm width. They are brown and are covered with silver hairs. They possess pappus with a length from 15 to 17 mm, which consists of 5–8 hairs stuck together at the base. The thousand seed weight without the calyx fluff is 3.60 ± 0.08 g.

As shown by the microscopic imaging, the mature cypsela (Figure 2A) was mostly occupied by a large embryo (Figure 2B–D), which was covered by the pericarp overlain by the testa (Figure 2E). In transverse section, the shape of the cypsela was mainly elliptical (Figure 2B). The embryo cells (cells of both the epidermis and parenchyma) were filled with storage materials in the form of approximately globular structures (Figure 2F). 

The NBB staining revealed that these structures were protein bodies (Figure 3A,B). The results of PAS reaction confirmed the presence of carbohydrates located mostly in the cell walls, and no starch grains were detected in the embryo cells (Figure 3C). The Sudan Black B staining gave a negative result, which indicated the absence of lipid bodies (Figure 3D). In the plant material treated with the osmium tetroxide solution (Figure 4A,B), some embryo cells had osmiophilic granules in the cytoplasm, suggesting their lipid character.

We did not detect starch grains in the embryo cells of *Carlina acaulis*; however, as shown by Jacobsen [19], the occurrence of starch grains in cotyledons depends on the species. In other species from the Asteraceae family starch was not found as storage material in embryos, e.g., in sunflower cultivars [15] and *Trichocline catharinensis* [20]. 

Protein reserves are an important storage of nitrogenous compounds, which are essential during germination and development of young seedlings. They are a common seed storage material [21]. In plants, proteins are selectively accumulated and packaged in specialized vacuoles referred to as protein bodies [22,23]. As reported by Boesewinkel and Bouman [21], protein bodies are located in the entire embryo or endosperm or they are restricted to a specialized layer. Here, we show that protein bodies in *Carlina acaulis* are found in the entire embryo and are dominant organelles in embryo cells. 

Studies have reported the presence of protein bodies in the embryo cells in the other members of the Asteraceae family as well [15,20]. 

Using chemical methods, we showed that *Carlina acaulis* cypselae contained oils (fatty acids); however, Sudan Black B staining gave a negative result, which indicated the absence of lipid bodies. In contrast, lipid bodies were a common cell feature of embryo cells in other Asteraceae species [15,20]. Thus, this problem deserves future analysis to be conducted using transmission electron microscopy.

Elias et al. [20] found that *Trichocline catharinensis* seeds were orthodox seeds. Additionally, our anatomical and histochemical observations indicate that *Carlina acaulis* seeds belong to this type.

### 2.2. Chemical Composition of the Cypsela

#### 2.2.1. Content of Soluble Proteins, Free Amino Acids, and Total Amino Acids

The microscopic analysis revealed the presence of protein bodies, which prompted quantification of protein and free amino acids. The *C. acaulis* cypsela was shown to contain 356.75 ± 30.45 and 3.38 ± 0.03 g/kg of soluble protein and free amino acids, respectively. Thus, the content of soluble protein in the *C. acaulis* cypsela is comparable to that in soybeans (360 to 460 g/kg), which is one of the richest vegetable sources of proteins [5]. The amount of proteins in the *C. acaulis* cypsela is also high in comparison with other Asteraceae plants. For example, the protein content in the *Helianthus annuus* L. cypsela is approx. 200 g/kg [24], and ranges from 250 to 300 g/kg in the cypsela of *Silybum marianum* (L.) Gaertner [25]. The relatively low content of free amino acids in *C. acaulis* cypselae is typical for plant seeds, e.g., in corn seeds, it ranges from 1 to 10% of the total pool of amino acids [26]. UPLC-FLD analysis showed the presence of eighteen amino acids, including nine essential, five conditionally essential, and four non-essential (Table 1). The total mass of the determined amino acids was 328.13 ± 40.10 g/kg cypselae, which corresponds to the soluble protein content.

#### 2.2.2. Oil Analysis

The oil obtained by hexane extraction of ground cypselae constituted 25 ± 2% of their weight. It was a light-yellow liquid (Figure 5B) absorbing UV-VIS radiation in the range of 190–450 nm (Figure 5A). The trace absorbance at 451 nm indicated the presence of small amounts of carotenoids, which was confirmed by the spectrophotometric analysis. It was shown that the content of carotenoids in the tested oil was 3.23 mg/g and the content of chlorophyll was 0.008 mg/g. Absorbance at a wavelength of 275 nm may indicate the presence of tocopherols, especially *alpha*-tocopherol absorbing radiation in the range of 265–310 nm (max 293 nm). The lack of absorbance at 320–370 wavelengths may indicate the absence of polyphenolic compounds such as chlorogenic acids, apigenin, quercetin, luteolin, and kaempferol derivatives.

The analysis of the oil carried out with the use of attenuated ATR-IR revealed the presence of ten peaks (Figure 5C). Abdullah et al. [27] recorded the IR spectrum of linolenic acid, which showed peaks with the following wavenumber values: 3009 cm^−1^ C=C bending vibration (aliphatic), 2927, 2855 cm^−1^ C–H stretching vibration (aliphatic), 1719 cm^−1^ C=O stretching vibration (carboxylic acid), 1284 cm^−1^ C–O stretching asymmetric (carboxylic acid), 937 cm^−1^ C–H bending vibration (alkene), and 722 cm^−1^ group vibration (aliphatic). The presence of some of these peaks (3008, 2922, 2852, 1743, and 721 cm^−1^) in the IR spectrum of the *C. acaulis* cypsela oil may suggest the presence of structural elements common to many fatty acids. Moreover, the wavenumber peaks 2922.48 and 2852.96 cm^−1^ can be derived from asymmetric and symmetric stretching vibrations of CH_2_ and CH_3_. Peaks with similar wavenumber values (2927 and 2868 cm^−1^) were identified by Guillen and Cabo [28] in the IR spectrum of tocopherol, which may confirm the presence of tocopherols in the tested oil. However, unambiguous assignment of specific structures to all recorded signals is difficult due to the complex qualitative composition and quantitative diversity of the sample components.

##### Fatty Acid Content

The analysis of the oil by GC-MS showed the presence of fifteen fatty acids, including ten unsaturated acids (Table 2). An example of a chromatogram and mass spectra of analytes are presented in Appendix A, respectively. 

Our study showed that the content of linolenic, 5-octadecenoic, stearic, palmitic, and oleic acids in the tested oil was 53.21, 18.66, 2.92, 7.18, and 4.94%, respectively, which is in agreement with the literature data [13]. Additionally, an unknown compound was visible in the chromatogram with a retention time of 13.3 min, which was probably cis-5-hexadecenoic acid, as suggested by the mass of the molecular ion and literature [13]. Our research showed that *C. acaulis* oil also contained myristic, palmitoleic, margaric, 8-octadecenoic, 9,12-octadecadienoic, arachidic, alpha-linolenic, 11,14,17-eicosatrienoic, and mangiferic acids and one unidentified compound (retention time 15.49 min). 

The differences between our findings and Spencer’s report [13] are probably the results of the use of different analytical techniques. It should be noted that Spencer et al. used gas chromatography with flame ionization detection and thin-layer chromatography in their research. Both these techniques carry a higher risk of misidentification of analytes, compared to the GC-MS used this study. Moreover, the analytical columns used in 1969 did not have such high resolution as those used today. This study therefore verifies the results of Spencer et al. and enriches the knowledge of the composition of *C. acaulis* cypsela oil with new data.

The high content of linoleic acid in the tested oil suggests its high pro-health value. Linoleic acid is a polyunsaturated fatty acid belonging to the omega-6 group. It takes part in the biosynthesis of physiologically important compounds such as prostaglandins, leukotrienes, and thromboxane and is a component of lipids found in cell membranes. For this reason, supplementation thereof in the diet is necessary for the proper functioning of the human organism [29].

##### Tocol Content

The chromatographic analysis showed the presence of three peaks corresponding to tocopherols. An unsymmetrical peak at a retention time of approximately 7 min indicated coelution of *gamma*- + *beta*-tocopherol. The dominant tocopherol in the *C. acaulis* oil was *alpha*-tocopherol (1.96 ± 0.09 g/kg oil). The content of *delta*- and *gamma*- + *beta*-tocopherols was significantly lower, i.e., 6.35 ± 0.41, and 39.20 ± 0.76 mg/kg, respectively. The chromatographic analysis did not reveal the presence of tocotrienols. An example of a chromatogram is presented in Appendix A. 

Vitamin E is a mixture of compounds with strong antioxidant properties that play an important role in the functioning of the human organism. These compounds must be supplied to the human organism with the diet. The main components of vitamin E are tocopherols, which protect low-density proteins and unsaturated fatty acids against oxidation [3]. The content of *alpha*-tocopherol in commonly produced vegetable oils has been shown to be approx. 120, 432, 173, and 260 (mg/kg) for canola, sunflower, corn, and soybean oils, respectively [30]. As a source of *alpha*-tocopherol, the *C. acaulis* cypsela oil is therefore over four-fold richer than sunflower oil and over sixteen-fold richer than canola oil. 

##### Pentacyclic Triterpenes

As shown in literature, pentacyclic triterpenes (ursolic and oleanolic acid, lupeol, amirin and their acetate derivatives) are present in the green parts and flower heads of *C. acaulis* [11]. Since these compounds are non-polar, they should also be present in the oil fraction [31]. Although they were found in seeds of many plants species [31,32,33], the chromatographic analysis revealed the absence of these triterpenes in the oil from *C. acaulis* cypselae.

#### 2.2.3. Mineral Composition

The analysis performed with the ICP-OES technique allowed determination of the profile of macro- and microelements in *C. acaulis* cypsela. The content of macronutrients (g/kg) was as follows: K 12.4 ± 0.07, Mg 5.20 ± 0.04, Ca 3.35 ± 0.03, P 12.04 ± 0.11, and S 7.82 ± 0.08. In turn, the following amounts of micronutrients (mg/kg) were detected: Cu 29.92 ± 0.24, Mn 18.19 ± 1.03, Mo 8.34 ± 0.08, Zn 81.04 ± 0.32, and Fe 109.70 ± 2.85. Co and Cr were not detected. Tyler and Zohlen [34] assessed the content of metals in seeds of 35 plant species growing on various types of soil. They showed the following ranges of the content (g/kg) of K, Mg, Ca, P, and S: 8.91–8.95, 2.31–2.53, 7.98–9.86, 4.15–7.71, and 2.44–2.47, respectively. They also showed that the contents of such micronutrients as Cu, Mn, Mo, Zn, and Fe were in the range of 11–12, 49–77, 0.12–0.15, 35–43, and 73 mg/kg, respectively. Załuski and Janeczko [35] reported that *Eleuterococcus senticosus* fruits contained the following amounts of macro- (g/kg) and micro- (mg/kg) elements: Ca 2.49–6.88, Mg 1.43–2.81, Fe 35–53, Mn 75–364, Zn 19–41, and Cu 3–13. Hence, it can be concluded that *C. acaulis* cypselae are a rich source of macro- and microelements, as the content of these components is on average two or three times higher than in seeds of other plants.

Some reports show that *C. acaulis* is a relatively good accumulator of metals, for example cadmium and silver [36,37]. Therefore, the content of toxic elements in cypselae was assessed. Pb and As were not detected; however, Cd (0.35 ± 0.00 mg/kg), Ni (2.51 ± 0.025 mg/kg), and Al (3.65 ± 0.02 mg/kg) were found. The contents of these metals were relatively low, which can be explained by the protective mechanism of plants preventing accumulation of heavy metals in generative organs [4].

#### 2.2.4. Polyphenol Content

The chromatographic analysis showed the presence of one dominant compound with a retention time of approx. 20 min, which was identified as chlorogenic acid. In addition, small amounts of neochlorogenic acid and trace amounts of more polar unidentified metabolites were visible in the chromatogram in the first 10 min of the analysis. An example of a chromatogram is presented in Figure 6. The content of neochlorogenic and chlorogenic acids was 0.22 ± 0.02 g/kg and 21.47 ± 1.36 g/kg, respectively. It should be emphasized that the content of chlorogenic acid in the *C. acaulis* cypsela is very high in comparison with other plants from the Asteraceae family. For example, sunflower seeds contain about 0.73–0.97 g/kg of this acid [38]. It is interesting that chlorogenic acid in the green parts of *C. acaulis* was at the level of 10 g/kg [12], which is more than two-fold lower than in the cypsela. In turn, 3,5-di-caffeoylquinic acid determined in the green parts [12] was not detected in the cypsela.

Chlorogenic acid is a well-investigated compound. Its multidirectional biological activity has been described in numerous scientific reports. It has been shown to have antioxidant, anti-inflammatory, antilipidemic, antidiabetic, and antihypertensive effects [39,40,41] and, therefore, it is a desirable component of plant-derived products.

Some reports described the presence of flavonoid compounds in *C. acaulis* plants [10], whereas the HPTLC screening of the hydrolyzed extract did not reveal the presence of flavonoid aglycones (myricetin, quercetin, apigenin, and kaempferol). This may indicate the absence of glycoside derivatives of these flavonoids in *C. acaulis* cypselae.

#### 2.2.5. Content of Volatile Compounds

The percentage composition of volatile compounds (VOC) detected in the *C. acaulis* cypselae is specified in Table 3. Examples of chromatograms and mass spectra are shown in Appendix A, respectively. Thirty-eight compounds belonging to different chemical classes were identified: ketones, aldehydes, esters, and heterocyclic hydrocarbons. The VOC composition in the seeds highly corresponded with their lipid content. Myristic and palmitic acid esters were contained in all the analyzed samples with relatively highest abundance in the cypselae. Monoterpenes and butanoic acid derivatives were present mainly in the ground seeds. The husk contained the lowest amounts of VOCs.

The predominant VOCs of the *C. acaulis* cypselae were the degradation products of higher fatty acids, obtained probably via enzymatic or oxidative hydrolysis of chemical bonds. 2- and 3-methylbutanoic acids are common compounds associated with fatty acid metabolism or lipid peroxidation. 2-Pentylfuran and hexanal are known products of linoleic acid oxidation while nonane, decane, limonene, and dodecane derive from oleic acid [42]. 

Monoterpenes, represented by eucalyptol, camphor, bornyl acetate, limonene, camphene, and 3-carene, were identified in *C. acaulis* embryo after grinding. They constituted about 15% of total VOCs, although they were not detected in the cypselae. They are probably located in the inner layer of cypsela idioblast cells and are available after crushing. Moreover, the husk constituted a barrier in the release of volatile compounds. Although terpenes are not predominant compounds in cypselae, they were reported in related species of the *Asteraceae* family, e.g., *Cirsium setidens* [43] or *Heliantum annuum* [24]. Interestingly, most of the monoterpenes observed in the study derive from one biosynthetic pathway, hence the low complexity of these derivatives.

## 3. Materials and Methods

### 3.1. Plant Material

Mature *C. acaulis* L. cypselae were obtained from the Botanical Garden of Maria Curie-Skłodowska University (UMCS) in Lublin, Poland (51°16′ N, 22°30′ E). They were identified by Mykhaylo Chernetskyy and deposited in the Botanical Garden of UMCS (voucher specimen no. 2005A). They were grown in an open field. The cypselae were collected in the second half of August 2019. Once harvested, the cypselae were dried at room temperature.

### 3.2. Light Microscopy

The cypselae were placed in a humid chamber for 2 h and later fixed in a mixture of 2.5% glutaraldehyde with 2.5% formaldehyde in 0.05 M cacodylate buffer (Sigma-Aldrich, Sigma-Aldrich Sp. z o.o. Poznan, Poland; pH 7.2). Next, they were washed three times in 0.1 M sodium cacodylate buffer, dehydrated in a graded ethanol series for 15 min at each concentration, and kept overnight in absolute ethanol. Subsequently, the samples were infiltrated for 1 h each in 3:1, 1:1, and 1:3 (*v*/*v*) mixtures of absolute ethanol and Technovit and then stored for 12 h in pure Technovit. The resin was polymerized by adding a hardener. The materials were cut into 5 μm thick sections using a rotary microtome, stained with 0.1% toluidine blue O, and mounted in DPX (Sigma-Aldrich, Sigma-Aldrich Sp. z o.o. Poznan, Poland). Selected Technovit sections were stained with NBB for total protein staining [44] or the PAS reaction was performed for visualization of starch and other insoluble polysaccharides [45]. Sudan Black B was used to detect the presence of lipids [46,47]. The sections were examined using a Nikon Eclipse E400 light microscope. A part of the material was fixed in a 1% osmium tetroxide solution (in cacodylate buffer, Sigma-Aldrich, Sigma-Aldrich LLB, Poznan, Poland; pH 7.2) at room temperature for 1.5 h. Next, this material was embedded in Technovit 7100 (Kulzer, Germany) for further histological analysis.

### 3.3. Scanning Electron Microscopy

For SEM, the cypselae were sputter-coated with gold and examined at an accelerating voltage of 20 kV using a Hitachi S-4700 scanning electron microscope (Hitachi, Tokyo, Japan), which is housed in the Institute of Geological Sciences, Jagiellonian University in Kraków, Poland.

### 3.4. Reference Standards and Chemicals

δ-tocopherol, (+)-γ-tocopherol, (±)-α-tocopherol, *D-α-tocotrienol*, *D*-δ-tocotrienol, γ-tocotrienol (analytical standards), methyl palmitate (≥99.0%), methyl stearate (≥99.5%), methyl oleate (≥99.0%), methyl linoleate (≥98.5%), methyl linolenate (≥99.0%), betulinic (≥98%), oleanolic (≥97%), ursolic (≥90%) acids, α-amyrin (≥98%), β-amyrin (≥98.5%), lupeol (≥94%), chlorogenic (≥95%), neochlorogenic (≥98%) acids, kaempferol (≥90%), apigenin (≥97%), myricetin (≥96%), quercetin (≥95%), 2-propanol (99.9%), hexane (≥95%), gradient grade acetonitrile, methanol. and ethanol, trifluoroacetic acid (≥99%), and trimethylsulfonium hydroxide (TMSH) (0.25 M methanolic solution) were purchased from Sigma Aldrich (St. Louis, MO, USA). Lupeol acetate (≥98%), α-amyrin acetate (≥98%), and β-amyrin acetate (≥98%) were provided by ALB Technology (ALB Materials Inc, Henderson, NV, USA). Water for HPLC was purified by ULTRAPURE Millipore Direct-Q^®^ 3UV-R (Merck Millipore, Billerica, MA, USA). Tert-butyl methyl ether (TBME) (99.8%) was purchased from Avantor Performance Materials Poland S.A. (Gliwice, Poland).

### 3.5. Determination of the Content of Protein, Free Amino Acids, and Total Amino Acids 

The protein content was assessed using a Bradford Assay [48,49]. Free amino acids were assessed using a ninhydrin assay [50]. The profile and total content of amino acids after acidic hydrolysis of cypselae were examined with the UPLC-PDA technique, in accordance with the methodology published by Szkudzińska et al. [51].

### 3.6. Isolation of Oil and Preparation of Samples for GC, HPLC, and HPTLC Analysis

The cypselae were dried, pulverized and accurately weighted (1.000 g). Samples were extracted four times with hexane (4 × 30 mL) using an ultrasonic bath (4 × 15 min.) The extracts were combined and evaporated in a rotary evaporator. Accurately weighed oil (approx. 100 mg) was transferred into a volumetric flask (5 mL) and dissolved in 2-propanol. The solution was filtered through a 0.25 µm polyamide membrane filter before the HPLC analysis. Accurately weighed oil (approx. 10 ± 2 mg) was dissolved in 500 μL TBME and derivatized by the addition of 250 μL TMSH. The whole sample was shaken vigorously and the GC analysis was performed.

For the determination of phenolic compounds, after extraction with hexane, the plant material was dried and extracted with 60% ethanol (3 × 3 mL) in an ultrasonic bath (3 × 15 min). The extracts were combined in volumetric flasks (10 mL) and made up to the mark with 60% ethanol. Prior to the HPLC analysis, the extracts were filtered through 0.22 µm nylon filters. Acidic hydrolysis of the extract was performed to determine the presence of flavonoid aglycones. A volume of 0.4 mL of 1.0 mol/L hydrochloric acid was added to 5 mL of the extract and the mixture was made up to 10 mL with methanol in a volumetric flask. Hydrolysis was conducted for 2 h at 40 °C. Before the analysis, the extract was neutralized with NH_3_ ∙ H_2_O.

### 3.7. Spectroscopic Analysis of Oil

The total chlorophyll and carotenoid contents were assessed with the method proposed by Wellburn [52]. The oil (50 mg) was mixed with 1 mL of isopropanol, vortexed for 5 min at 3000 rpm, and centrifuged at 6000 rpm. Spectrophotometric measurements were carried out at 470, 646, and 663 nm for carotene, chlorophyll b, and chlorophyll a, respectively. The DAD spectrum of the oil was recorded using a VWR Hitachi Chromaster 600 chromatograph with a 5430 Diode Array Detector and EZChrom Elite software (Merck, Darmstadt, Germany). The ATR-IR spectroscopy analysis was performed according to the procedure published previously [53].

### 3.8. Fatty Acid Analysis

The analysis was performed using an Agilent GC-MSD system (GC/MSD 6890N/5975) equipped with a HP-88 Agilent capillary column (60 m × 0.25 mm; 0.20 μm film thickness), MSD ChemStation ver. E.02.02.1431 software (Agilent Technologies, Santa Clara, CA, United States), and a split–splitless injector. The oven temperature was programmed from 110 °C to 190 °C with 8 °C/min, hold for 2 min at 110 °C and 13 min at 190 °C. The temperature of the injector was 250 °C. The injection volume was 1 μL (split ratio 150:1; split flow 180 mL/min). Helium was used as a carrier gas at a flow rate of 1.2 mL/min. A quadrupole mass spectrometer with electron ionization (EI) at 70 eV and with a full scan type acquisition mode (50 *m*/*z* to 500 *m*/*z*) was used as a detector connected with the GC. The temperature of the MS source and the MS quadrupole was set to 230 °C and 150 °C, respectively. The identification of the constituents was based on a comparison of their mass spectra with the mass spectra library NIST resources and retention times with standards. Each sample was measured in three replicates.

### 3.9. HPLC and HPTLC Analysis

The analysis was performed on a VWR Hitachi Chromaster 600 chromatograph with a 5430 Diode Array Detector, a 5440 FL Detector, and EZChrom Elite software (Merck, Darmstadt, Germany). An RP18e LiChrospher 100 column (Merck, Darmstadt, Germany) (25 cm × 4.0 mm i.d., 5 µm particle size) was used for the analyses. The identity of compounds was established by comparison of retention times and PDA spectra with the corresponding standards.

Chlorogenic acids were determined using an isocratic system. The mobile phase consisted of acetonitrile-water-trifluoroacetic acid (8:92:0.025 *v*/*v*/*v*). The eluent flow rate was 1 mL/min. The column temperature was set to 25 °C. The injection volume was 1 μL. Data were collected between 190 and 400 nm. The quantitative analysis was performed at λ = 321 and 324 nm for neochlorogenic and chlorogenic acids, respectively. Each sample was measured in three replicates.

Tocopherols were determined using an isocratic system. The mobile phase consisted of acetonitrile and methanol (5:95 *v*/*v*). The eluent flow rate was 1.2 mL/min. The column temperature was set to 30 °C. The injection volume was 5 μL. The quantitative analysis was performed using a fluorescence detector with an excitation wavelength at 296 nm and an emission wavelength at 330 nm. Each sample was measured in three replicates.

The presence of pentacyclic triterpenes was assessed using previously published methods [11]. The presence of flavonoid aglycones was assessed by HPTLC and a previously published procedure [54].

### 3.10. Determination of Minerals

Determination of minerals was conducted according to the methodology published previously (Dresler et al., 2020). Pulverized cypselae (0.2000 g) were mineralized using 5 mL of the HNO_3_: H_2_O_2_ mixture (4:1 *v*/*v*) in DigiPREP (SCP Science, Clark Graham Baie D’Urfé, QC, Canada). The process was carried out at a temperature of 120 °C for 2 h. The resulting clear solutions were transferred into volumetric flasks and filled up to 25 mL with deionized water. The amounts of all elements were measured using ICP-OES PlasmaQuant PQ 9000 Elite (Analityk Jena AG, Jena, Germany). Effective plasma power was 1300 W and the plasma, auxiliary, and nebulizer argon flow rates were 14.0, 0.5, and 0.6 L/min, respectively. The attenuated radial direction of measurement for Ca, K, and Mg (analytical lines: 315.887, 766.491, and 285.213, respectively) and the axial direction for Cd, Zn, Mn, Fe, Mo, Cu, Co, Cr, Ni, Al, Pb, and As (analytical lines: 214.441, 206.200, 257.610, 259.940, 202.030, 2019.227, 228.615, 205.552, 231.604, 396.152, 220.353, and 188.970, respectively) were applied. Each sample was measured in three replicates.

### 3.11. Volatile Compound Analysis

The analyses of the *C. acaulis* cypselae were performed on a whole embryo with the husk (=cypsela), husk, and a peeled and ground embryo. Extraction was carried out using 50/30 µm DVB/CAR/PDMS SPME fiber (Supelco, Bellefonte, PA, USA). 2-undecanone (Merck, Poland) was used as an internal standard—1.0 or 0.2 mg/mL of water for the seeds and husk, respectively. A sample containing 100 mg of cypselae and the embryo or 50 mg of husk was placed in a 4 mL vial capped with an aluminum cap. Equilibration was performed at 60 °C for 30 min. The fiber exposition time was 15 min, and thermal desorption was carried out for 3 min at 250 °C directly in the GC injection port. All analyses were performed in triplicate.

The analysis was performed using an Agilent 7890B GC coupled with the 7000GC/TQ system (Agilent Technologies, Paolo Alto, CA, USA). Separation was carried out on an HP-5 MS column; 30 m × 0.25 mm × 0.25 µm (J&W, Agilent Technologies, Palo Alto, CA, USA) at a constant helium flow of 1 mL/min. The injector temperature was set at 250 °C and the sample was applied in a split mode (20:1). The temperature program was 50 °C for 1 min, followed by 4 °C/min to 130 °C, 10 °C/min to 280 °C, and held isothermal for 2 min. The MS source was set at 230 °C, the transfer line was 320 °C, and the quadrupole temperature was 150 °C. The electron ionization energy was set at 70 eV, scan range, m/z 30–400.

The identification was performed using MassHunter Workstation Software, Version B.08.00 coupled with the NIST17 mass spectra library, and accomplished by comparison with RI. The RI of the compounds were calculated using a series of n-alkanes (C8-C20, Merck, Poland).

## 4. Conclusions

In this study, the morphological and anatomical features as well as the chemical composition of *C. acaulis* cypselae were described. Our research showed that the cypselae are a rich source of nutrients such as protein (with a high proportion of essential amino acids), unsaturated fatty acids, and macro- and microelements; therefore, they can be a valuable supplement of the diet. The high content of tocopherols and chlorogenic acid indicates the health promoting properties of the cypselae, as the biological activity of these compounds, including strong antioxidant, anti-inflammatory, antidiabetic, antilipidemic, and antihypertensive activity, is widely described in the scientific literature. This study expands the knowledge of *C. acaulis.*

## Figures and Tables

**Figure 1 ijms-21-09230-f001:**
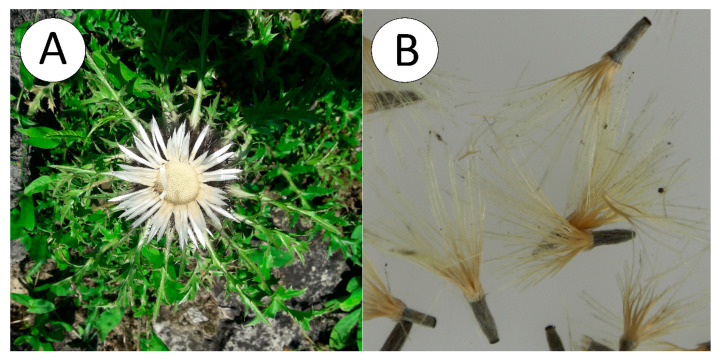
(**A**). Flowering plant *Carlina acaulis* L., (**B**). Mature cypselae.

**Figure 2 ijms-21-09230-f002:**
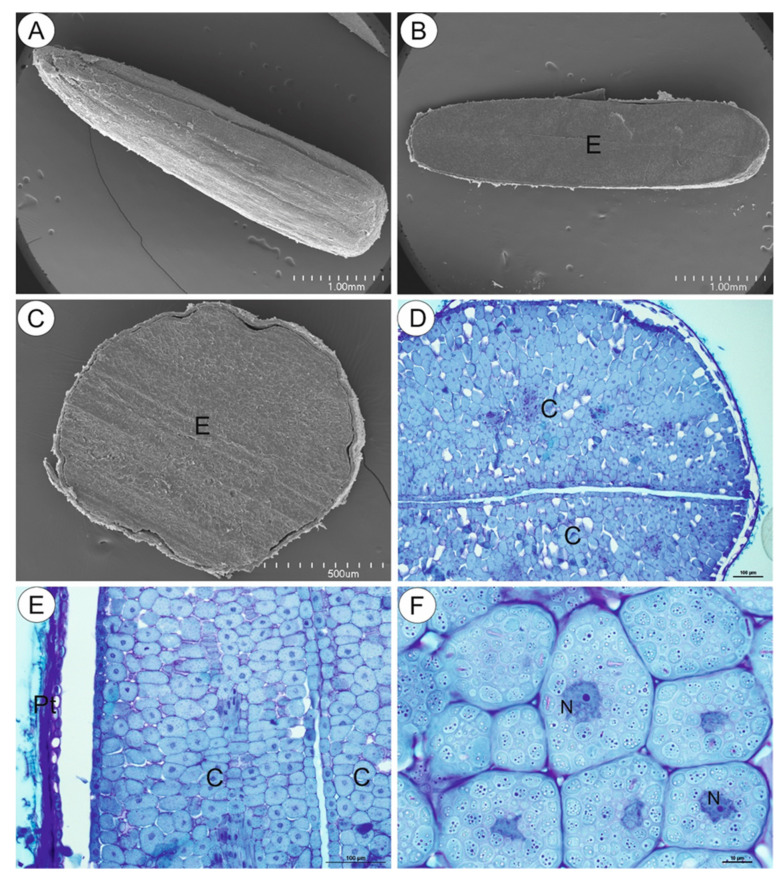
Structure of the *Carlina acaulis* cypsela. (**A**). Morphology of the cypsela in scanning electron microscopy (SEM), bar 1 mm. (**B**). Transverse section of the cypsela; embryo (**E**) in SEM, bar 1 mm. (**C**). Cross section through the cypsela; embryo (**E**) in SEM, bar 500 µm. (**D**). A part of cross section through the cypsela; two cotyledons are well visible (**C**), bar 100 µm. (**E**). A part of transverse section through the cypsela; pericarp and testa (Pt), cotyledon (**C**), bar 100 µm. (**F**). Parenchyma cells of the cotyledon; note the cytoplasm filled with storage material, nucleus (N), bar 10 µm.

**Figure 3 ijms-21-09230-f003:**
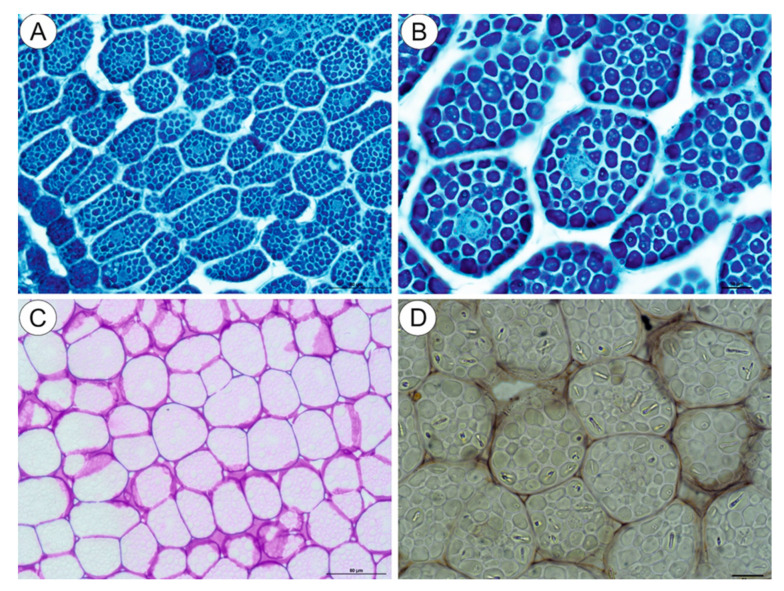
Cytochemistry of *Carlina acaulis* embryo cells. (**A**,**B**). NBB staining, bar 50 µm and bar 10 µm. (**C**). PAS reaction’ note the presence of polysaccharides only in the cell walls, bar 50 µm. (**D**). Sudan Black B staining, bar 10 µm.

**Figure 4 ijms-21-09230-f004:**
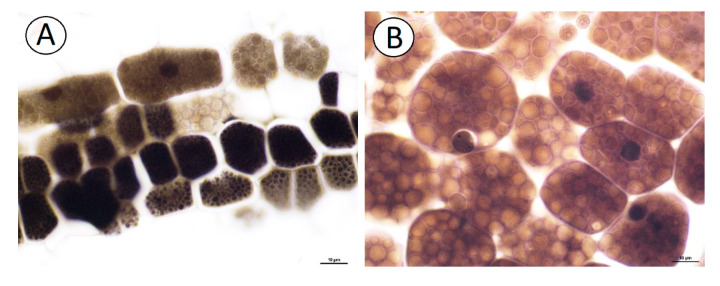
Cytochemistry of *Carlina acaulis* embryo cells. (**A**). Material treated with the osmium tetroxide solution; some embryo cells have osmiophilic granules in the cytoplasm, bar 10 µm. (**B**). Material treated with the osmium tetroxide solution. Embryo cells; note the cytoplasm filled with protein bodies that are not osmiophilic, bar 10 µm.

**Figure 5 ijms-21-09230-f005:**
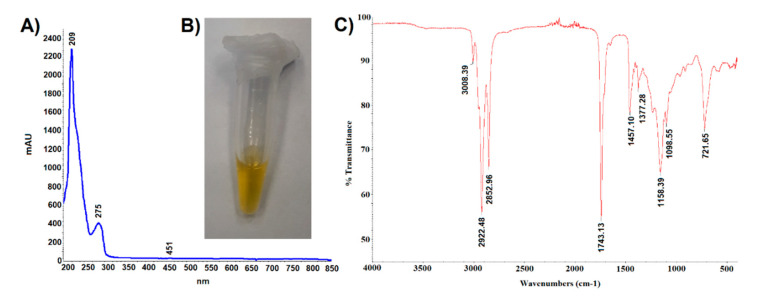
(**A**). UV-VIS spectroscopic fingerprint, (**B**). photography, and (**C**). infrared spectroscopic fingerprint of *Carlina acaulis* cypsela oil.

**Figure 6 ijms-21-09230-f006:**
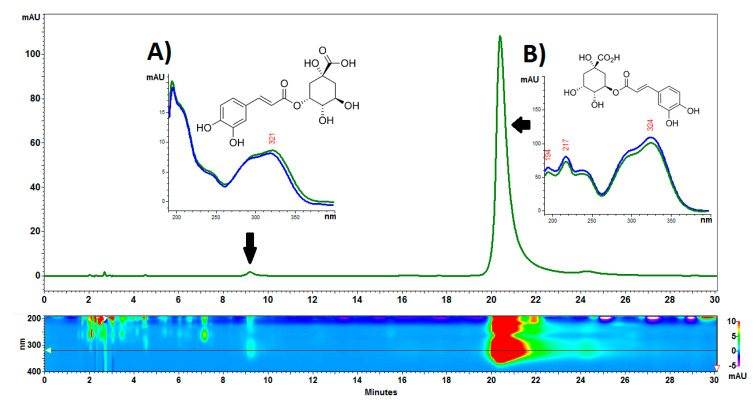
Example of an HPLC-DAD chromatogram of *Carlina acaulis* L. cypsela extract. (**A**). neochlorogenic acid, (**B**). chlorogenic acid. The analysis was performed on an RP18e LiChrospher 100 column (25 cm × 4.0 mm i.d., 5 µm particle size). The column temperature was set to 25 °C. Acetonitrile-water-trifluoroacetic acid (8:92:0.025 *v*/*v*/*v*) at a flow rate of 1 mL/min was used as eluent.

**Table 1 ijms-21-09230-t001:** Content of amino acids (g/kg ± SD) in *Carlina acaulis* cypsela (n = 3).

Essential Amino Acid		Conditionally Essential		Non-Essential Amino Acids	
Compound Name	Content	Compound Name	Content	Compound Name	Content
Histidine	9.69 ± 1.26	Arginine	37.4 ± 4.86	Alanine	14.2 ± 1.99
Isoleucine	9.65 ± 1.16	Cystine	7.91 ± 1.03	Aspartic acid	32.2 ± 4.19
Leucine	22.7 ± 2.50	Glycine	20.3 ± 1.62	Glutamic acid	79.5 ± 10.3
Lysine	9.30 ± 0.84	Proline	11.8 ± 1.06	Serine	15.7 ± 2.04
Methionine	6.18 ± 0.87	Tyrosine	9.25 ± 1.30		
Phenylalanine	14.0 ± 1.82				
Threonine	11.3 ± 1.46				
Tryptophan	3.55 ± 0.32				
Valine	13.5 ± 1.48				

**Table 2 ijms-21-09230-t002:** Content of fatty acids (% m/m ± SD) in *Carlina acaulis* cypsela oil (n = 3).

Compound Name	FA Abbreviation	RT FAME (min)	Mass Data FAME	Content of FA
Myristic acid	C14:0	11.09 ± 0.1	242(M), 211, 199, 185, 166, 157, 143, 129, 115, 97, 87, 83, 74, 55	0.07 ± 0.10 *
Palmitic acid	C16:0	12.98 ± 0.1	270(M), 255, 239, 227, 213, 199, 185, 171, 157, 143, 129, 115, 97, 87, 74, 69, 55	7.18 ± 0.20 *
cis-5-hexadecenoic acid	C16:1 (*n*-11)	13.33 ± 0.1	283, 268(M), 250, 236, 219, 194, 165, 152, 123, 110, 96, 84, 74, 67, 55	0.86 ± 0.10 ^
Margaric acid	C17:0	13.99 ± 0.1	313, 284(M), 253, 241, 199, 143, 101, 87, 74, 69, 55	0.05 ± 0.10 ^
Stearic acid	C18:0	15.13 ± 0.1	298(M), 267, 255, 241, 199, 157, 143, 129, 111, 97, 87, 74, 69, 55	2.92 ± 0.05 *
Unidentified compound	-	15.49 ± 0.2	264, 247, 222, 207, 180, 166, 137, 110, 96, 84, 74, 67, 55	0.16 ± 0.10 ^
5-Octadecenoic acid	C18:1 (*n*-13)	15.58 ± 0.2	296(M), 278, 264, 247, 235, 222, 207, 194, 180, 166, 152, 137, 123, 110, 96, 84, 74, 67, 55	18.66 ± 0.03 ^
Oleic acid	C18:1 (*n*-9)	15.81 ± 0.2	313, 296(M), 278, 264, 235, 222, 193, 180, 166, 110, 97, 83, 69, 55	4.94 ± 0.15 *
8-Octadecenoic acid ^a^	C18:1 (*n*-10)	15.92 ± 0.2	313, 296(M), 264, 235, 222, 207, 180, 166, 138, 123, 111, 97, 83, 69, 55	0.32 ± 0.00 ^
9,12-Octadecadienoic acid	C18:2 (*n*-6)	16.34 ± 0.2	294(M), 263, 220, 150, 95, 67, 55	0.09 ± 0.00 *
Linoleic acid	C18:2 (*n*-6)	17.04 ± 0.2	294(M), 263, 245, 233, 220, 205, 191, 178, 164, 150, 135, 123, 109, 95, 81, 67, 59, 55	53.2 ± 1.07 *
Arachidic acid	C20:0	17.96 ± 0.2	326(M), 295, 283, 269, 255, 241, 227, 199, 185, 171, 149, 143, 137, 129, 115, 97, 87, 83, 74, 69, 55	0.83 ± 0.05 *
alpha-Linolenic acid	C18:3 (*n*-3)	18.62 ± 0.2	292(M), 250, 208, 179, 163, 149, 121, 108, 95, 79, 67, 55	0.58 ± 0.00 *
11,14,17-Eicosadienoic acid ^b^	C20:3 (*n*-3)	18.98 ± 0.2	292, 250, 208, 179, 163, 149, 121, 108, 95, 79, 67, 55	0.12 ± 0.00 ^
Mangiferic acid	C18:2 (*n*-3)	19.30 ± 0.2	294(M), 263, 193, 164, 150, 123, 109, 95, 81, 67, 55	0.09 ± 0.00 ^

RT—retention time, FAME—Fatty Acid Methyl Esters, FA—Fatty Acid, M—molecular ion, ^a^—or 11-Octadecenoic acid, ^b^—or 9,12-Octadecadienoic acid, *—calculated on the basis of a calibration curve, ^—calculated with the internal normalization method.

**Table 3 ijms-21-09230-t003:** Percentage of volatile compounds identified in *Carlina acaulis* cypsela (% ± SD).

RI Calc.	RI Lit.	RT (min)	Compound	Formula	Crushed Embryo *	Husk **	Cypsela *
802	801	4.33	Hexanal	C_6_H_12_O		6.94 ± 2.00	0.64 ± 0.12
837	834	5.32	3-Methylbutanoic acid	C_5_H_10_O_2_	15.85 ± 8.26		
841	841	5.41	2-Methylbutanoic acid	C_5_H_10_O_2_	0.58 ± 0.14		
854	846	5.55	2-Hexenal	C_6_H_10_O		0.09 ± 0.07	2.57 ± 2.14
878	866	6.22	Pentanoic acid	C_5_H_10_O_2_	0.65 ± 2.45		0.02 ± 0.04
895	901	6.75	Heptanal	C_7_H_14_O		0.76 ± 0.14	0.01 ± 0.02
930	929	7.62	α-Thujene	C_10_H_16_	2.59 ± 0.76	0.40 ± 0.22	
946	943	8.03	Camphene	C_10_H_16_	1.85 ± 0.72	0.42 ± 0.11	
954	959	8.24	γ-Valerolactone	C_5_H_8_O_2_	0.83 ± 0.11		
982	973	9.02	Hexanoic acid	C_6_H_12_O_2_	3.16 ± 0.77	0.63 ± 0.07	2.56 ± 0.28
991	993	9.31	2-Pentylfurane	C_9_H_14_O	2.18 ± 0.70	0.28 ± 0.12	0.06 ± 0.02
1003	1004	9.64	Octanal	C_8_H_16_O		1.59 ± 0.16	0.93 ± 0.30
1030	1029	10.38	D-Limonene	C_10_H_16_	1.03 ± 0.28		
1033	1031	10.47	Eucalyptol	C_10_H_18_O	10.41 ± 2.53	1.73 ± 0.68	
1060	n.d.	11.25	4,5-Dimethylnonane	C_11_H_24_	9.67 ± 3.07	0.35 ± 0.12	0.31 ± 0.02
1104	1105	12.65	Nonanal	C_9_H_18_O	1.48 ± 0.34	13.94 ± 2.06	2.18 ± 0.50
1120	1123	13.05	2-Ethylhexanoic acid	C_8_H_16_O_2_	2.25 ± 1.23		0.45 ± 0.08
1148	1146	13.83	Camphor	C_10_H_16_O	4.16 ± 1.52	1.42 ± 0.38	
1162	1164	14.25	2-Nonenal	C_9_H_16_O		5.11 ± 0.97	
1174	1175	14.60	Octanoic acid	C_8_H_16_O_2_		1.15 ± 0.18	1.51 ± 0.97
1206	1206	15.58	Decanal	C_10_H_20_O	4.44 ± 1.17	15.26 ± 2.39	8.56 ± 3.72
1271	1272	17.32	Nonanoic acid	C_9_H_18_O_2_			2.09 ± 0.59
1288	1285	17.81	Bornyl acetate	C_12_H_20_O	2.97 ± 0.87	2.79 ± 0.82	
1294	1294	17.98	2-Undecanone	C_11_H_22_O	1.01 ± 0.48	13.43 ± 9.92	17.09 ± 3.29
1308	1308	18.32	Undecanal	C_11_H_22_O		1.48 ± 0.07	0.69 ± 0.18
1399	1400	20.33	Tetradecane	C_14_H_30_		0.73 ± 0.45	0.30 ± 0.07
1409	1405	20.53	Dodecanal	C_12_H_24_O		0.76 ± 0.03	1.01 ± 0.23
1456	1455	21.38	trans-Geranylacetone	C_13_H_22_O	1.46 ± 1.23	2.00 ± 0.36	3.95 ± 1.35
1468	1462	21.61	Alloaromadendrene	C_15_H_24_	1.63 ± 0.43	3.47 ± 0.96	0.12 ± 0.10
1514	1513	22.44	Tridecanal	C_14_H_22_O		3.67 ± 0.85	
1826	1826	27.08	Isopropyl myristate	C_17_H_34_O_2_	2.50 ± 0.30	1.41 ± 0.87	30.94 ± 1.96
1843	n.d.	27.30	Farnesyl acetaldehyde	C_17_H_28_O	2.12 ± 1.20	1.06 ± 0.42	3.89 ± 1.96
1874	n.d.	27.69	Phthalic acid, hept-4-yl isobutyl ester	C_19_H_28_O_4_	1.04 ± 0.66	1.01 ± 0.50	1.83 ± 0.10
1898	1900	28.01	Nonadecane	C_19_H_40_	1.64 ± 0.28	0.61 ± 0.19	2.60 ± 0.53
1964	n.d.	28.77	2-Ethylhexyl octadecyl carbonate	C_17_H_34_O_3_	7.39 ± 0.53	3.30 ± 0.27	9.45 ± 1.53
2024	2023	29.44	Isopropyl palmitate	C_19_H_38_O_2_	4.49 ± 0.38	1.87 ± 0.23	5.35 ± 0.70
2098	2100	30.21	Heneicosane	C_21_H_44_			0.42 ± 0.07
2173	n.d.	30.95	2-Ethylhexyl 4-methoxycinnamate	C_18_H_26_O_3_	0.45 ± 0.07	0.14 ± 0.04	

RT—retention time, RI calc—retention index, calculated, RI lit—retention index, literature, n.d.—no data. Internal standard—2-Undecanone—added in * 1.0 µg and ** 0.2 µg. All analyses were performed in triplicate.

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
