# Peer review of "Morphological, Anatomical, and Phytochemical Studies of Carlina acaulis L. Cypsela"

_ijms, 2020, doi:10.3390/ijms21239230_

Round 1

Reviewer 1 Report

In this article, the authors carried out a morphological, anatomical, and phytochemical study of Carlina acaulis L. cypsela seeds. The results obtained are of high interest to society and the science community, reason because I consider that this article should be accepted. However, I have detected some lack of information that is important to include in the article, reason because I suggest this article with minor revisions.

Introduction: I consider the Introduction section lack information, especially on two important points.

First, the authors talk about the seed oils and some of the components that have been detected in its composition. However, the authors should include one paragraph about the uses and applications of these components on medicine, agriculture, or pharmaceutical between others. In order to remark the importance of knowing the Carlina acaulis L. cypsela seeds composition in order to obtain an enriched extract with multiple applications.

On the other hand, authors should mention the analytical techniques used for the root and green parts analysis and the analytical technique employed for seed analysis in 1969, with the aim of readers could understand the differences and advantages/disadvantages of the current techniques used.

Results:

-Table 1: please homogenize the format of this table, some of the titles are in bold letters and others not.

-Table 2: "cis-5-hexadecenoic acid ?" please delete the "?".

-In general, the authors use "±" with space before and after but on some occasions you delete them, please review them in order to homogenize the format.

-In section "2.2.3" C. acaulis is mentioned two times but an italic letter is not used, please correct it.

-Line 212: "3.5-di-caffeoylquinic", please correct to "3,5-di-caffeoylquinic".

Conclusions: please, include a conclusion section in the article.

References: please review the format of references 3, 19, 36, and 37.

Author Response

Dear Reviewer,

Thank you very much for your time and valuable comments, which all have been considered and incorporated. The detailed list of responses is given below. The changes made in the revised manuscript are marked in red. We hope that the modifications and explanation will be acceptable for you.

Yours sincerely,

Strzemski, corresponding author

November 23, 2020

In Lublin, Poland

Reviewer

In this article, the authors carried out a morphological, anatomical, and phytochemical study of Carlina acaulis L. cypsela seeds. The results obtained are of high interest to society and the science community, reason because I consider that this article should be accepted. However, I have detected some lack of information that is important to include in the article, reason because I suggest this article with minor revisions.

Introduction: I consider the Introduction section lack information, especially on two important points.

First, the authors talk about the seed oils and some of the components that have been detected in its composition. However, the authors should include one paragraph about the uses and applications of these components on medicine, agriculture, or pharmaceutical between others. In order to remark the importance of knowing the Carlina acaulis L. cypsela seeds composition in order to obtain an enriched extract with multiple applications.

On the other hand, authors should mention the analytical techniques used for the root and green parts analysis and the analytical technique employed for seed analysis in 1969, with the aim of readers could understand the differences and advantages/disadvantages of the current techniques used.

REPLY: Dear Reviewer, thank you very much for your positive opinion. The necessary supplementing information pointed by you has been introduced to the manuscript in the Introduction and section 2.2.2.2.

Results:

-Table 1: please homogenize the format of this table, some of the titles are in bold letters and others not.

REPLY: The formatting has been standardized.

-Table 2: "cis-5-hexadecenoic acid ?" please delete the "?".

REPLY: Removed.

-In general, the authors use "±" with space before and after but on some occasions you delete them, please review them in order to homogenize the format.

REPLY: The formatting has been standardized.

-In section "2.2.3" C. acaulis is mentioned two times but an italic letter is not used, please correct it.

REPLY: This has been corrected.

-Line 212: "3.5-di-caffeoylquinic", please correct to "3,5-di-caffeoylquinic".

REPLY: This has been corrected.

Conclusions: please, include a conclusion section in the article.

REPLY: The "Conclusions" section has been added

References: please review the format of references 3, 19, 36, and 37.

REPLY: The References have been formatted according to journal requirements.

Reviewer 2 Report

The manuscript presented by the authors fits the IJMS scope.

The article has a simple design, and some methods are outdated.

The aim of the study should be rephrased (rows 65-68). The discussion section is missing for the macro and microscopy analysis and should be significantly improved for section 2.2 Chemical composition of the cypsela.

What is the purpose of UV-VIS and IR analysis? In my opinion, the method for chlorophyll and carotenoids is inappropriate and should not be used.

Please add a conclusion section and highlight the novelty of the study!

Minor

Row 28 – please correct the error

Please use italics for C. acaulis.

Author Response

Dear Reviewer,

Thank you very much for your time and valuable comments, which all have been considered and incorporated. The detailed list of responses is given below. The changes made in the revised manuscript are marked in red. We hope that the modifications and explanation will be acceptable for you.

Yours sincerely,

Strzemski, corresponding author

November 23, 2020

In Lublin, Poland

Reviewer

The manuscript presented by the authors fits the IJMS scope.

The article has a simple design, and some methods are outdated.

REPLY: Dear Reviewer. Thank you for the evaluation of our manuscript. We hope you will be satisfied with the changes we have introduced. Most of the techniques used in our study are modern (HPLC-DAD-FLD, GC-MS, ICP-OES). Spectrophotometry and HPTLC were used only for general test and rapid screening.

The aim of the study should be rephrased (rows 65-68). The discussion section is missing for the macro and microscopy analysis and should be significantly improved for section 2.2 Chemical composition of the cypsela.

REPLY: The aim of the study has been re-edited. Discussion in the microscopy analysis section has been added. We have also expanded the discussion in section 2.2.

What is the purpose of UV-VIS and IR analysis? In my opinion, the method for chlorophyll and carotenoids is inappropriate and should not be used.

REPLY: These methods allowed collecting general data on the tested oil. The content of carotenoids and chlorophylls in vegetable oils is very often given in publications and the spectroscopic technique is commonly used for this purpose. The UV and IR spectra may be a valuable fingerprint of the test oil and may suggest the presence or absence of certain metabolites. The presented data are a supplementation to the analyses described in the manuscript. However, if the Reviewer deems it necessary, we will transfer them to the supplementary material.

Please add a conclusion section and highlight the novelty of the study!

REPLY: The "Conclusions" section has been added.

Row 28 – please correct the error

REPLY: This has been corrected.

Please use italics for C. acaulis.

REPLY: This has been corrected.

Round 2

Reviewer 2 Report

The article has been improved significantly and it’s suitable for publication in IJMS.